# In Vivo *Piggy*bac-Based Gene Delivery towards Murine Pancreatic Parenchyma Confers Sustained Expression of Gene of Interest

**DOI:** 10.3390/ijms20133116

**Published:** 2019-06-26

**Authors:** Masahiro Sato, Emi Inada, Issei Saitoh, Shingo Nakamura, Satoshi Watanabe

**Affiliations:** 1Section of Gene Expression Regulation, Frontier Science Research Center, Kagoshima University, Kagoshima 890-8544, Japan; 2Department of Pediatric Dentistry, Graduate School of Medical and Dental Sciences, Kagoshima University, Kagoshima 890-8544, Japan; 3Division of Pediatric Dentistry, Department of Oral Health Sciences, Course for Oral Life Science, Graduate School of Medical and Dental Sciences, Niigata University, Niigata 951-8514, Japan; 4Division of Biomedical Engineering, National Defense Medical College Research Institute, Saitama 359-8513, Japan; 5Animal Genome Unit, Institute of Livestock and Grassland Science, National Agriculture and Food Research Organization (NARO), 2 Ikenodai, Tsukuba, Ibaraki 305-0901, Japan

**Keywords:** pancreas, *piggy*Bac transposon, in vivo electroporation, intraparenchymal injection, gene delivery, persisted expression of transgenes, Splinkerette-PCR, chromosomal integration

## Abstract

The pancreas is a glandular organ that functions in the digestive system and endocrine system of vertebrates. The most common disorders involving the pancreas are diabetes, pancreatitis, and pancreatic cancer. In vivo gene delivery targeting the pancreas is important for preventing or curing such diseases and for exploring the biological function of genes involved in the pathogenesis of these diseases. Our previous experiments demonstrated that adult murine pancreatic cells can be efficiently transfected by exogenous plasmid DNA following intraparenchymal injection and subsequent in vivo electroporation using tweezer-type electrodes. Unfortunately, the induced gene expression was transient. Transposon-based gene delivery, such as that facilitated by *piggy*Bac (PB), is known to confer stable integration of a gene of interest (GOI) into host chromosomes, resulting in sustained expression of the GOI. In this study, we investigated the use of the PB transposon system to achieve stable gene expression when transferred into murine pancreatic cells using the above-mentioned technique. Expression of the GOI (coding for fluorescent protein) continued for at least 1.5 months post-gene delivery. Splinkerette-PCR-based analysis revealed the presence of the consensus sequence TTAA at the junctional portion between host chromosomes and the transgenes; however, this was not observed in all samples. This plasmid-based PB transposon system enables constitutive expression of the GOI in pancreas for potential therapeutic and biological applications.

## 1. Introduction

The pancreas is an important organ that functions in both the digestive and endocrine systems in mammals. Functional defects in this organ are associated with the manifestation of several serious disorders, such as diabetes, pancreatitis, and pancreatic cancer [1]. Gene delivery targeting the pancreas appears to be a promising tool for curing such disorders and for exploring the mechanisms underlying these complex diseases. Unfortunately, few studies exist that examine in vivo gene delivery targeted to the pancreas. For example, some groups employed non-viral vectors that are encapsulated by liposomes [2] or complexed with ployvinylpyrrolidone (PVP) [3]. Other groups utilized viral vectors such as adenoviruses [4,5,6,7,8], adeno-associated viruses (AAV) [9,10,11], and lentiviruses [12]. All of these reports employed systems where DNA was injected intraperitoneally, intravenously, or intraductally to transfect pancreatic cells in adult mice, and although they achieved successful transfection of pancreatic cells (including acini and islet cells), the level and distribution of gene products varied among those reports. Intraparenhymal injection of DNA is provides another route to deliver exogenous DNA to pancreatic cells [3,13,14], and this method appears to be simpler than the other gene delivery routes mentioned above as it does not require any specific skill or other surgery-related equipment. For this method, relatively large amounts of DNA solution (100–120 μL) containing liposomally encapsulated plasmid DNA are injected with the aid of a 33- or 27-gauge needle [3,13]. As a result, successful transfection of acinar and β cells and other tissues was achieved, but the expression level of the associated genes appears to be low, likely due to diffusion of the injected solution into an interstitial space among the pancreatic cells prior to being trapped by cells. Houbracken et al. [14] microinjected solutions containing lentiviral or adenoviral vectors directly into the pancreatic parenchyma at multiple sites and demonstrated that a high efficiency could be obtained by adenoviral (but not lentiviral) vectors, resulting in transient transduction of mainly exocrine acinar cells. Even with these attempts, however, successful long-term gene expression in pancreatic tissues after gene delivery has not been achieved, likely due to the use of plasmids that are difficult to integrate into the host genome.

We employed a novel gene delivery system targeting the pancreas that is called intra-pancreatic parenchymal injection for gene transfer (IPPIGT) [15], and this system enables DNA to be efficiently delivered inside the pancreatic cells due to adequate electric fields provided by an electroporator. In vivo electroporation (EP) using electroporators has been used as a powerful tool to delivery DNA to the target cells, tissues, and organs of mammals [16]. For IPPIGT, we aspirate only 1-2 μL of DNA solution containing circular plasmid DNA and dye (trypan blue; TB) by a mouth-piece-controlled micropipette, and then we inject the solution into the inner pancreatic parenchyma with the aid of a dissecting microscope. The injected portion that is easily recognized by the presence of dye is next electroporated in vivo using the tweezer-type electrodes [15]. As a result, we found focal and distinct expression of a gene introduced at the injected site when inspection was performed 1 day after IPPIGT. Unfortunately, we found that gene expression of the introduced DNA did not continue over a week, likely due to the transient nature of the plasmid itself. To explore the effects of gene products on pancreatic function over a long period of time, it is vital that the exogenous DNA construct becomes integrated into the host chromosomes. 

*Piggy*Bac (PB) is a commercially available transposon-based gene delivery system [17,18,19] that has been employed for a variety of applications including in vitro transfection in various mammalian cells [20,21,22,23,24,25,26,27], generation of genome-edited cells [28,29,30] and inducible pluripotent stem (iPS) cells [31,32,33,34,35], generation of transgenic animals [36,37,38,39,40,41], in vivo gene transfer in mice [42,43,44,45,46,47,48,49,50,51], and gene discovery via insertional mutagenesis [52,53,54]. It is also a useful tool for obtaining stable transfectants from a small number (5.7 × 10^4^) of hard-to-transfect cells [55]. The PB-based gene delivery is very simple, as researchers can easily add a PB transposase expression vector and transposon vectors carrying a gene of interest (GOI) flanked by the two inverted terminal repeats (ITR) sequences. When they are placed inside a cell, transposase binds to the ITR to allow the GOI to be individually integrated into host chromosomal sites that contain the TTAA sequence, which is duplicated on the two flanks of the integrated fragment [56,57].

In this study, we tested if this PB system confers persistent (at least over 1 month) expression of a GOI in murine pancreas after IPPIGT has been performed. Once optimized, PB-based transposon systems can be used to test the possibility of converting abnormal phenotypes associated with inherited pancreatic disorders, such as diabetes, by the continuous expression of therapeutic genes.

## 2. Results and Discussion

We initially examined if the PB-based gene delivery confers long-term expression of a GOI when IPPIGT was performed using adult murine pancreas. For this, we used a PB transposon vector (pT-tdTomato) carrying a cDNA encoding red fluorescent protein (tandem dimer Tomato, tdTomato) (as GOI) and a PB transposase expression vector (pTrans) to test gene transfer efficiency in vivo (Figure 1A). On 1 day following IPPIGT, strong red fluorescence was observed in the unfixed samples from both the experimental (co-injection with pTrans + pT-tdTomato) and the control group (injection with pT-tdTomato alone) (a,b vs. c,d in the left panel of Figure 2A). The intensity decreased rapidly, however, when samples were inspected 2 weeks after IPPIGT, although fluorescence in the experimental group was more extensive than that in the control group (e,f vs. g,h in the left panel of Figure 2A). At 1.5 months post IPPIGT, nearly all of the fluorescence disappeared from the control sample, while distinct and condensed fluorescence was still detected in the experimental group (i,j vs. k,l in the left panel of Figure 2A). Quantification of tdTomato-derived fluorescence in samples after IPPIGT demonstrated that there was a significant difference between the experimental and control groups in the levels of fluorescence when samples were dissected 1.5 months after IPPIGT (right panel of Figure 2A). These findings suggest that PB is beneficial for guaranteeing continuous gene expression of a GOI in adult pancreas.

As PB is known to exert efficient chromosomal integration of transposons [58], we next assessed the GOI in the DNA-injected pancreatic cells using molecular biological methods. We first examined the presence of the GOI (tdTomato expression unit) in the genomic DNA isolated from both experimental and control pancreas by PCR (Figure 2B). As expected, the GOI was detectable in the samples isolated from the experimental group on 1 day, 2 weeks, and 1.5 months after IPPIGT. In contrast, it was difficult to detect the GOI in the control samples examined after 2 weeks and beyond. These results suggest that the GOI may have been integrated into the genome of some pancreatic cells after the PB-based gene delivery.

To confirm the above results at the histological level, we next performed cryostat sectioning of the dissected pancreas containing the injected site, which is easily recognizable by the presence of TB and fluorescence (as shown in the left panel of Figure 2B). In this case, we employed pT-EGFP, an enhanced green fluorescent protein (EGFP) expression transposon vector (Figure 1A), instead of pT-tdTomato for IPPIGT, as non-specific fluorescence was high when cryostat sections of pancreas transfected with pTrans + pT-tdTomato were inspected (data not shown). Inspection of green fluorescence in the pancreas 1.5 months after IPPIGT with pTrans and pT-EGFP revealed distinct fluorescence in the experimental group but not in the control groups (IPPIGT with pT-EGFP alone) (a,b vs. f,g in Figure 3A), which was consistent with the previous results (i,j in the left panel of Figure 2A) obtained with the use of pT-tdTomato as a transposon. Inspection of cryostat sections revealed that the fluorescent acini in the experimental group were more abundant than those observed in the control group (c-e vs. h-j in Figure 3A). These findings support the above conclusion of a beneficial role for PB in enabling persistent expression of a GOI.

It is established that PB-based integration events are preceded by a junction between the ITR and TTAA on the chromosomes [19,59]. We therefore examined this possibility using Spinkerette-PCR [60], a useful method to assess the junctional sequence between host chromosome and the transgene. The pancreas (injected with pTrans + pT-EGFP) was removed 12 days or 1.5 months after IPPIGT. Next, the injected portion that is identifiable by the presence of TB and EGFP-derived fluorescence was dissected using microscissors under a fluorescence dissecting microscope. The upper panels of Figure 3B-a,b, illustrate a portion of the pancreas dissected 12 days after IPPIGT that exhibits distinct green fluorescence (arrowed in Figure 3B-b). Similarly, pancreas sampled 1.5 months after IPPIGT (shown in the upper panels of Fig. 3B-c,d) exhibited fluorescence, although its strength appears to be weak (arrowed in Figure 3B-d) compared to the fluorescence intensity observed in that of the samples sampled 12 days after IPPIGT (b vs. d in Figure 3B). The dissected fluorescent pancreas was then subjected to genomic DNA isolation prior to Splinkerette-PCR and subsequent subcloning into a TA cloning vector. A total of 12 clones for each group (sampled on 12 days or 1.5 months after IPPIGT) were isolated and subjected to PCR using universal primers (M2 and RV) to amplify the insert. In the lower panels of Figure 3B, a portion of data obtained through 2% agarose gel electrophoresis is shown. There are several clones possessing variously-sized inserts (marked by asterisks; ranging from ~250 to ~800 bp). Direct sequencing of these PCR products using the universal primers demonstrated that portion of the clones (enclosed by circles on each lane) were successfully incorporated into the mouse genome sequence; however, other clones possessed irrelevant sequences, likely due to unknown contamination generated during cloning process. The summary of sequencing results is provided in Table 1. In both groups, there are clones carrying portions of the mouse genome sequence that were derived from each different chromosome (e.g., chromosomes 1, 2, 5, 7, 8, 9, 10, 14, 16, and Y). Notably, of the 7 clones derived from the pancreas sampled 12 days after IPPIGT, only 3 (43%) were identified as authentic clones possessing the desired consensus sequence (TTAA) and subsequent host (murine) chromosomal sequence (Table 1). Similar events were also observed in the samples examined 1.5 months after IPPIGT. Specifically, of the 6 clones, only 1 (17%) exhibited TTAA at the 5’ end of the inserts.

It has been widely established that the PB transposon system is useful for efficient integration of GOIs into host chromosomes in cultured cells and for efficient transgenesis in various animals [18]. However, only few studies have been available whether this system is also effective in vivo. For example, Saridey et al. [42] demonstrated that a single injection of plasmid-based PB transposons via tail vein confers persistent expression of a GOI (coding for luciferase) in the liver and lungs of mice, suggesting chromosomal integration of GOI. Similar results were also provided by other groups who used repeated intravenous injections of PB transposons [43] or intravenous injection of hybrid PB/viral vectors [48]. PB transposons have also been used to create experimental systems for exploring gene therapeutic strategy. For example, Cheng’s group modified a hydrodynamics-based gene delivery (HGD) system that is successfully targeted to kidneys in live mice, and they demonstrated that HGD using PB transposons carrying either glutathione S-transferase A4 [61] or insulin-like growth factor-1 receptor [62] protected against obstruction-induced renal fibrosis. Viral and nonviral gene transfer vectors as well as cell-based therapies are currently under investigation as tools for correction of hemophilia A, caused by a deficiency in factor VIII (FVIII) [63]. For example, more recently, investigators have demonstrated phenotypic correction of hemophilia A [64,65] or B [66] after PB-mediated gene transfer to mouse liver. One report coupled the use of PB with a cell-type-specific promoter to improve efficiency of gene expression in vivo [65,66]. Others have created hybrid adenovirus and AAV PB vectors capable of in vivo gene transfer [48,49]. Such hybrid or cell-type-specific vectors can overcome the limitations of delivering plasmid DNA into tissues by harnessing viruses to gain entry into the desired cell-type of interest.

The most advantageous property of using a PB system enabling chromosomal integration of a GOI is the ability to allow for the persistent expression of a GOI. For example, continuous gene expression in vivo was achieved for ~300 days in liver [42], ~80 days in liver [43], ~6 months in liver [44,45], and ~35 weeks in nasal airways [48]. Additionally, expression for ~6 months in tails [50] and 56 days in liver [51] was achieved after the tail vein-mediated gene delivery. Consistent with those reports, we successfully achieved persistent (at least 1.5 months after gene delivery) expression of GOI through direct gene delivery of PB transposons to pancreatic parenchyma. Taken together, our results indicate that the PB-based gene delivery system is very useful for allowing persistent expression of a GOI, even in vivo.

It has been demonstrated that the PB-based integration of a GOI requires the presence of a TTAA recognition sequence in endogenous genes of a host [19]. In our present study, however, we found that 57% and 83% of pancreas sampled on 12 days or 1.5 months after IPPIGT exhibited chromosomal integration of the GOI into the noncanonical (i.e., not TTAA) site (Table 1). Notably, rare integration of PB into noncanonical (i.e., not TTAA) sites has been previously observed [67]. Li et al. [68] recently characterized a genome-code insertion site preference of PB by sequencing a large set of integration sites arising from PB gene delivery using mouse embryonic stem (ES) cells, and they demonstrated that PB can integrate at noncanonical integration sites (a site other than its known TTAA insertion) at a low frequency (2%). According to Li et al. [68], PB can insert into sequences other than TTAA that are predominantly CTAA/TTAG and ATAA/TTAT, and the universal TA is thought to be conserved. In our study, the sequences we observed in the site recognized as the noncanonical site at the PB 5′ ITR-genomic junction appeared to be variable, and they included AAGC, ACAA, AATT, GATC (for 2 cases), GGGG, GGAT, TTGA, and AAGA (see Table 1). These listed sequences all appear to not always to follow the rule (TA is universal for the noncanonical sites recognized by the PB system) suggested by Li et al. [68]. As mentioned above, there are several reports of in vivo gene transfer experiments using the PB system, but to our knowledge, all these reports indicate that the PB transposon-mediated integration sites follow the consensus sequence TTAA [42]. In our previous paper [27] involving transfection of porcine primary cultured cells with a plasmid-based PB transposon system, all resulting clones possessed a TTAA consensus sequence at the integration site when the same Splinkerette-PCR approach used in this study was employed. While the sample size is too small to draw conclusions concerning integration patterns, these data suggest that PB-based chromosomal integration of a GOI employing a noncanonical site would occur frequently in vivo, as suggested by in vitro gene delivery studies using cultured cells. 

Our IPPIGT system enables overexpression of a target gene in pancreatic cells (mainly acinar cells) in vivo [15]. This technology is not only useful for continuous gene expression of a transgene when the PB gene delivery system is combined with IPPIGT (in this study) but also theoretically applicable to knock out or knock down of a target gene using a recently developed genome editing [as exemplified by clustered regularly interspaced short palindromic repeats/CRISPR associated proteins (CRISPR/Cas9)] and RNA interference (RNAi) technologies, respectively. These approaches will be valuable when IPPIGT is used for basic research towards future clinical applications.

In conclusion, we successfully created mice that exhibited continuous expression of a GOI within the pancreas by employing a PB transposon system coupled with IPPITG. These techniques will be useful for rapidly examining the function of a given GOI in vivo and for biomedical research examining gene therapy in the context of pancreatic disorders such as diabetes, pancreatitis, and pancreatic cancer.

## 3. Materials and Methods

### 3.1. Mice

Female B6C3F1 mice, 8–10 weeks of age (Kyudo Co., Ltd., Tosu, Saga, Japan), were used for IPPIGT. The mice were maintained on a 12 h light/dark cycle (lights on from 07:00 h to 19:00 h) and were provided with food and water ad libitum. The experiments described were performed in agreement with the guidelines of the Kagoshima University Committee on Recombinant DNA Security (No. 25076; dated on 27 March 2014), and they were based on the “Guide for the Care and Use of Laboratory Animals” of the National Academy of Sciences, USA. Additionally, they were approved by the Animal Care and Experimentation Committee of Kagoshima University (Sakuragaoka Campus) (No. MD14003; dated on 17 April 2014). The experiments describing in vivo transfection of mouse pancreas by IPPIGT were accompanied by surgery (exposure of spleen/pancreas) and operation/manipulation (DNA injection toward pancreatic parenchyma and in vivo EP). All efforts were made to minimize the number of animals used and their suffering.

### 3.2. Plasmid Construction

The plasmid vectors pTrans, pT-EGFP, and pT-tdTomato [26,27] used in this study are shown in Figure 1A. Briefly, pTrans is a vector allowing for the expression of the PB transposase under the chicken β-actin gene-based promoter CAG [69]. pT-EGFP is derived from a pPB-based vector that contains two PB acceptors with inverted repeats, and it carries an *EGFP* cDNA expression unit [CAG promoter + *EGFP* cDNA + poly(A) sites]. pT-tdTomato is a pPB-based vector that carries a *tdTomato* cDNA expression unit (kindly provided by Dr. Roger Tsien) under the control of the CAG promoter. All plasmids were grown in *Escherichia coli* DH5α and purified using a MACHEREY-NAGEL plasmid purification kit as described by Sato et al. [70]. 

For IPPIGT, these plasmids were dissolved in circular form in Dulbecco’s modified Ca^2+^, Mg^2+^-free phosphate-buffered saline (PBS) + 0.1% (v/v) TB (Trypan Blue Stain 0.4%; Invitrogen Co., Carlsbad, CA, USA; used to enable visual identification of the injected site) at a final concentration of 0.5 μg/μL for each DNA plasmid. As an experimental group, a mixture containing pTrans and pT-tdTomato or pTrans and pT-EGFP was used. As control group, pT-tdTomato or pT-EGFP alone was used.

### 3.3. IPPIGT

IPPIGT was performed according to the method described by Sato et al. [15]. Briefly, mice were anesthetized by intraperitoneal (IP) injection of three combined anesthetics (medetomidine, 0.75 mg/kg; Nippon Zenyaku Kogyo Co. Ltd., Fukushima, Japan), midazolam (4 mg/kg; Sandoz K.K., Tokyo, Japan), and butorphanol (5 mg/kg; Meiji Seika Pharma Co., Ltd., Tokyo, Japan)], and then both spleen and pancreas were exposed after making a small dorsal incision at the left side (Figure 1B-a). A 1 to 2 μL solution containing plasmid DNA and TB was injected into two different sites (both of which are close to each other) inside the pancreatic parenchyma by a glass pipette connected to a mouthpiece under inspection using a dissecting microscope (Figure 1B-b,f). After injection, the injected sites were covered with a small piece of wet paper (KimWipe; Jujo-Kimberly Co. Ltd., Tokyo, Japan) (Figure 1B-c,g), compressed between a pair of disc-type tweezer-type electrodes (with 7 mm of inner diameter; #CUY650P7; Nepa Gene Co., Ltd., Chiba, Japan) (Figure 1B-d,h) and then electroporated using the square-pulse generator NEPA21 (Nepa Gene Co., Ltd.) that generates two types of pulses, specifically the poring pulse (Pp) and the transfer pulse (Tp). In this study, in vivo EP was performed using the conditions that included 4 Pp (2.5 ms wavelength/50 ms duration/50V) and 8 Tp (50 ms wavelength/50 ms duration/20V) that were previously determined by Sato et al. [15]. After IPPIGT, the electroporated pancreas was returned to its original position, and the abdominal wound was closed. The anesthetized mice were recovered by subcutaneous injection of atipamezole (3.75 mg/kg; Nippon Zenyaku Kogyo Co. Ltd.), a medetomidine antagonist, and then warmed on an electric plate warmer. Sampling of the pancreas was performed at 1 day, 2 weeks, and 1.5 months post-IPPIGT as described in Figure 1C.

### 3.4. Observation of Fluorescence and Dissection of Tissues

At the indicated days after gene delivery, mice were sacrificed by cervical dislocation. The electroporated portion of the pancreas, which was easily identified by the presence of TB, was directly inspected for fluorescence under an Olympus SZX12 fluorescence stereomicroscope (Tokyo, Japan) and photographed using a digital camera (FUJIX HC-300/OL; Fuji Film, Tokyo, Japan). Images were then printed using a digital color printer (CP700DSA; Mitsubishi, Tokyo, Japan). In certain cases, the injected portion (approximately 3–5 mm^3^) of pancreas that was identifiable by the presence of residual co-injected TB beneath the parenchymal capsule and sometimes by fluorescence was dissected by microscissors (#MB-53; NAPOX; Natsume, Tokyo, Japan) and subjected to histological processing or genomic DNA isolation, as described below and in Figure 1C.

### 3.5. Data Analysis

In an attempt to measure the strength of tdTomato-derived fluorescence in the samples (shown in the left panel of Figure 2A) obtained after IPPIGT, the photographic data (corresponding to a 3 mm^2^ portion that had been injected with transposons and contained co-injected TB) were incorporated into a Macintosh computer and processed using a program set with Adobe Photoshop ver. 5 (Adobe System, Inc., Seattle, WA, USA) to quantify the levels of fluorescence. Data were obtained from three injected areas of a pancreas in mice that had been subjected to IPPIGT at different days for each group. The raw data were then normalized to the data obtained from another pancreatic area that had been untransfected. The data were then subjected to statistical analysis using GraphPad PRISM 5 for Windows software (GraphPad software, Inc., La Jolla, CA). Data were analyzed by one-way repeated ANOVA and expressed as mean ± SD. Statistical significance was determined by Student’s *t*-test. *p*-Values <0.05 were considered statistically significant.

### 3.6. Tissue Processing

A portion of the samples (approximately 3–5 mm^3^) dissected from the pancreas were fixed in 4% paraformaldehyde in PBS at 4 °C for 2 days, dehydrated by immersion in 0.25% sucrose in PBS at 4 °C for 2 days, and then dehydrated in 0.4% sucrose in PBS at 4 °C for 4 days. These samples were then embedded in O.C.T. compound [Tissue-Tek^®^ (no. 4583); Miles Scientific, Naperville, IL, USA] for cryostat sectioning. The sections were then embedded in a solution containing glycerol and 600 nM 4′,6-diamidino-2-phenylindole (DAPI; Molecular Probes Inc. Eugene, OR) for 5 min at room temperature, inspected for fluorescence using an Olympus BX60 microscope, and photographed using a Mitsubishi digital color printer.

### 3.7. PCR

Genomic DNA was extracted by adding 300 μL of lysis buffer [0.125 μg/mL of proteinase K, 0.125 μg/mL of Pronase E, 0.32 M sucrose, 10 mM Tris-HCl (pH 7.5), 5 mM MgCl_2_, and 1% (v/v) Triton X-100] to a section of pancreas (~3 mm^3^) in a 1.5-mL Eppendorf tube and then incubating for 2–3 days at 37 °C, followed by phenol/chloroform extraction. Next, the supernatant was isopropanol-precipitated. The precipitated DNA was dissolved in 6 μL of sterile water and stored at 4 °C. The resulting genomic DNA (1 μL; ~5 ng) were subjected to PCR using the primer set TDR-3S (5′-CCC GTA ATG CAG AAG AAG-3′) and TDR-3RV (5′-GTG ATG TCC AGC TTG GTG TCC-3′), which yields 206 bp products from the middle region of *tdTomato* cDNA (accession no. AY678269). The final PCR volume was 20 μL. The primer set mEx4-2S (5′-TGAATCGAGCAGGTGTTTCAT-3′)/mEx4-2RV (5′-AGGAACACAGGAAGACTGGAC-3′) was used for detection of endogenous mouse α-1,3-galactosyltransferase (*α-GalT*) gene as a reference. This primer set yields 344-bp fragments [71]. As a negative control, 0.5 μg of genomic DNA from intact mouse tails was used. For positive controls, 5 ng of pT-tdTomato plasmid was used. PCR was performed under thermocycler conditions that included initial denaturation (92 °C for 10 min), followed by 40 cycles of denaturation (96 °C for 10 s), annealing (56 °C for 1 min), and polymerization (72 °C for 2 min), and a final extension at 72 °C for 5 min using Taq polymerase (TaKaRa Taq; #R001A, Takara Shuzo, Tokyo, Japan). The PCR products (2 μL) were separated on a 2% agarose gel and visualized by staining with ethidium bromide.

### 3.8. Genomic Integration Site Analysis Using Splinkerette-PCR

Splinkerette-PCR was performed to map PB integration sites within the DNA-introduced pancreas according to the method of Potter and Luo [60] and based on the manufacturer protocol [72]. Sau3AI-digested genomic DNA isolated from the pancreatic tissues of the in vivo electroporated mice was ligated using a Splinkerette adapter generated by annealing HMSpAa and HMSpBb, and junction fragments were PCR amplified using primers HMSp1 and PB-R-Sp1. Nested PCR was performed using primers HMSp2 and PB-L-Sp2. PCR products were cloned into the TA-cloning vector pCR2.1 (Invitrogen Co.). The recombinant colonies were then subjected to PCR using universal primers [M2 (5′-CCCAGTCACGACGTT-3′) and RV (5′-CAGGAAACAGCTATGAC-3′)] (accession no. MK176932.1) specific for the plasmid. The resulting PCR products were then electrophoresed in a 2% agarose gel to estimate the size of each PCR product. The PCR products containing over 300 bp were subjected to sequencing with the M2 or RV primer. True PB integration sites were considered if the genomic sequence began immediately after the terminal TTAA at the end of the 3′ ITR sequence and matched at the genomic locus with 93% identity.

## Figures and Tables

**Figure 1 ijms-20-03116-f001:**
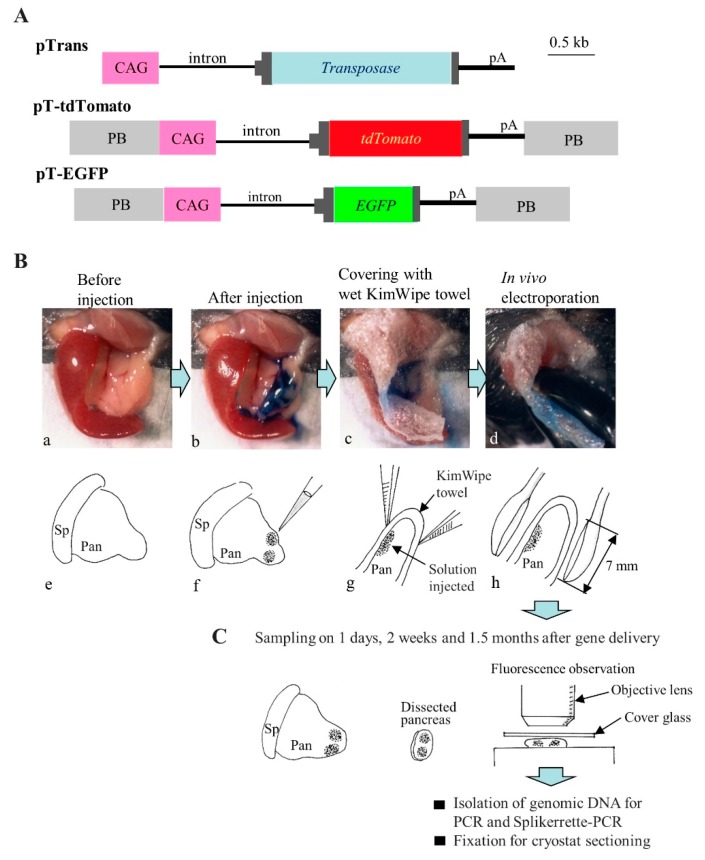
(**A**) Schematic representation of the *piggy*Bac-based transposon vectors used in this study. The plasmid backbone is not shown in the figure. CAG, cytomegalovirus enhancer + chicken β-actin promoter; pA, poly(A) sites; *EGFP*, enhanced green fluorescent protein cDNA; PB, acceptor site in the *piggy*Bac system; *tdTomato*, Tandem dimer Tomato cDNA; *Transposase*, PB transposase gene. (**B**) The IPPIGT procedure illustrated by photographs (**a**–**d**) and schematic (**e**–**h**). Spleen (Sp) and pancreas (Pan) were exposed after dorsal incision of the skin and muscle wall under anesthesia (**a**,**e**). Pancreatic parenchyma was injected with a small volume of solution (1–2 μL) containing plasmid DNA and TB (**b**,**f**). Then, the injected site of the pancreatic parenchyma was covered with wet paper (**c**,**g**). Immediately after covering, the injected site was compressed between the two tweezer-type electrodes and electroporated (**d**,**h**). At 1 day, 2 weeks, and 1.5 months post-IPPIGT, sampling was performed. (**C**) A process for dissection of the injected site in pancreas, and subsequent histological and molecular biological analyses. Following IPPIGT, the spleen (Sp) and pancreas (Pan) are exposed (**a**). Then, pancreas containing the injected site, which is easily recognized by the presence of residual TB (**b**) and sometimes fluorescence (**c**), is dissected and used for analysis.

**Figure 2 ijms-20-03116-f002:**
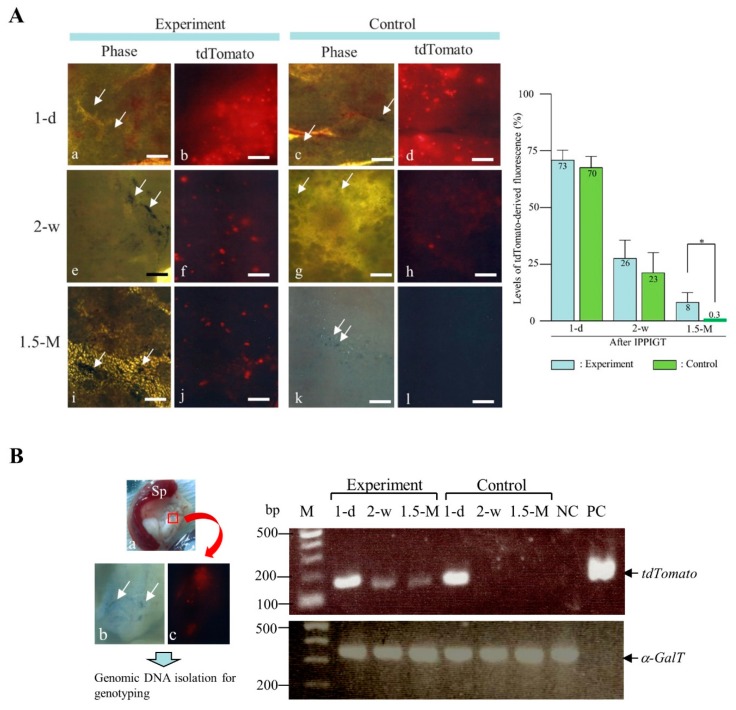
Inspection of the tdTomato-derived fluorescence and PCR analysis of pancreas dissected at 1 day (1-d), 2-w (2 weeks), and 1.5-M (1.5 months) post-IPPIGT with pTrans + pT-tdTomato (Experiment) or pT-tdTomato alone (Control). (**A**) Fluorescence in intact samples (left panel). Arrows indicate the presence of TB that marks the sites where DNA was injected. Phase, photographs taken under white light; tdTomato, photographs taken under white light + UV. Bar = 1 mm. Quantification of tdTomato-derived fluorescence in samples after IPPIGT (right panel). The mean fluorescence level ± SD was graphically represented. The number shown at the top of each column indicates the mean fluorescence level (%). * represents *p* < 0.02. (**B**) PCR analysis of pancreatic tissue. Transposons in the experimental or control group shown in (**A**) were subjected to IPPIGT, and the injected portion (enclosed by quadrant shown in a of the left panel), which is visible by the presence of TB (shown by arrows in b) and shows tdTomato-derived fluorescence (shown in b,c of the left panel), was isolated 1-d, 2-w and 1.5-M post-IPPIGT for genomic DNA isolation. Samples from each group were then subjected to PCR analysis (right panel). The same samples were also subjected to PCR using primers to detect the endogenous α-GalT gene (*GGTA1*). M, 100-bp ladder markers; lane NC, intact murine tail DNA; lane PC, control plasmid DNA (pT-tdTomato for detection of *tdTomato* cDNA).

**Figure 3 ijms-20-03116-f003:**
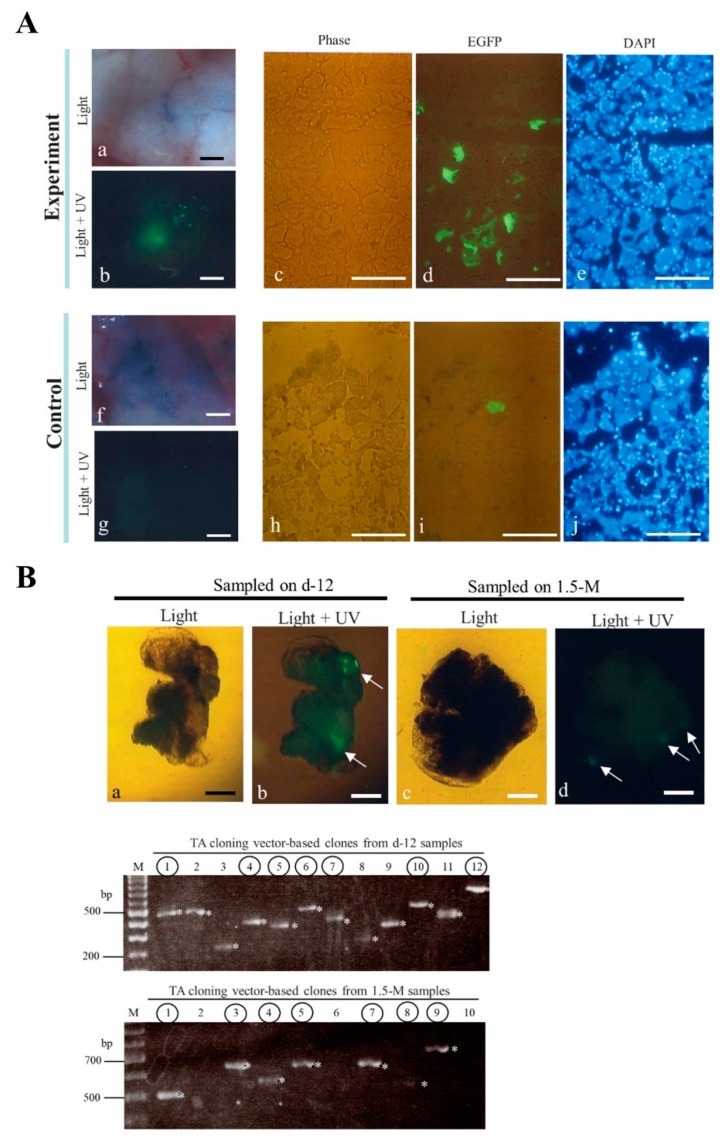
Inspection of the EGFP-derived fluorescence in the pancreas 1.5 months after IPPIGT (upper panels) and Splinkerette-PCR analysis (lower panels). (**A**) Adult pancreas 1.5 months after IPPIGT with pTrans + pT-EGFP (Experiment) or pT-EGFP alone (Control). The injected portion (which can be visualized by the presence of TB and expression of green fluorescence shown in a,b and f,g) was then subjected to cryostat sectioning. The sections were inspected for EGFP fluorescence after counterstaining with DAPI (c–e for experimental group and h–j for control group). Light, photographs taken under light; Light + UV, photographs taken under light + UV. Bar = 100 μm. (**B**) Dissection of pancreas 12 days (12-d) or 1.5 months (1.5-M) after IPPIGT with pTrans + pT-EGFP (upper panels) and 2% gel electrophoresis of Splinkerette-PCR products (lower panels). A portion of pancreas showing EGFP-derived fluorescence (shown by arrows in a–d) was dissected under inspection using a fluorescence dissecting microscope. Genomic DNA isolated from the dissected pancreatic tissues was next subjected to Splinkerette-PCR as described in the materials and methods section. The resulting Splinkerette-PCR products were then cloned into TA cloning vectors. PCR was performed on the resulting TA clones as a template using universal primers, and the products were analyzed in 2% agarose gels (lower panel). Light, photographs taken under light; Light + UV, photographs taken under light + UV. Bar = 100 μm.

**Table 1 ijms-20-03116-t001:** Splinkerette-PCR analysis.

Samples ^1^	Sequence (5′–3′) ^2^	Reference ^3^
12-d-1	**AGCAGT**TTAAAGCAGGGACTGCTCTTGGCAGTGGACCCACCCTGCACAGTACACAGGCTCCTGTCCGGTA	AC148000| *Mus musculus* BAC clone RP24-105F6 from chromosome y, complete sequence; Identities = 64/64 (100%)
12-d-2	**ATTGAC**AAGCACGCTCAATGTCGAGCCCCAATCCCTCCAACGTTTCTCTTGATCCCA	AC153511| *Mus musculus* 10 BAC RP23-102N12 (Roswell Park Cancer Institute (C57BL/6JFemale) Mouse BAC Library) complete sequence; Identities = 50/51 (98%)
12-d-3	**GCATTG**ACAAGCACGCATACACATACATGCACACATGCACCT	AL513354| Mouse DNA sequence from clone RP23-150J22 on chromosome 15; Identities = 35/36 (97%)
12-d-4	**GGTTTC**AATTTCTTTAGTATATTCAAGCTCCGTTACCAGAGACAACTTTGGAATACAGCATCTCA	AC162174| *Mus musculus* chromosome 1, clone RP23-306P24, complete sequence; Identities = 59/59 (100%)
12-d-5	**ACGCCT**TTAATCCCAGCACTCGGGAGGCAGAGACAGGCAGATTTCTGAGTTCGAGGTCAGCC	CT030190| Mouse DNA sequence from clone RP23-115D20 on chromosome 16; Identities = 56/56 (100%)
12-d-6	**TAGTGG**GATCCTGGCTGTCTAGACATGTACATGATGGGCACCAAGTAAACAAGATTGATAATGAGGAAGCAGAGCTGAATAATGAAAGACACCTCGCAAAATGGGGCAT	AC132104| *Mus musculus* BAC clone RP24-364N7 from chromosome 9, complete sequence; Identities = 103/103 (100%)
12-d-7	**GAAACT**TTAAGCCCTGAAAAATTGCCTGTCAGCTTGTACCCATAAAGTAGTCTGTATAT	AC153650| *Mus musculus* BAC clone RP24-93I18 from chromosome 1, complete sequence; Identities = 189/189 (100%)
1.5-M-1	**TAGTGG**GATCGCTTGGATTCCCAGAATCCCAGTTATCTCTCCCTGCTGTCTGGGTATCCCGT	AC125181| *Mus musculus* BAC clone RP23-324L6 from chromosome 8, complete sequence; Identities = 56/56 (100%)
1.5-M-2	**GAGACC**GGGGTAAGTATGACAGTCCCCAGTGTGTGCCCTTGACTGGAATGCAGGGGTGGTGAGATGGAGCGGGATGAGGAGGAGATGGCTGAGCCTCATGTGTGGAAAGCAAGGATGCAAACAGTACCCCAC	AC142244|*Mus musculus* chromosome 1, clone RP23-79H24, complete sequence; Identities = 126/126 (100%)
1.5-M-3	**CTAGTG**GGATCCCCACCAACACATAATAGCCAGGGGCAGCAGTATATCTATATCTCCTGCAGTGGTGTATGTGGGGGG	AC157809|*Mus musculus* chromosome 1, clone RP24-375B12, complete sequence; Identities = 70/71 (98%)
1.5-M-4	**GAGACC**TTGAATCTTGTTCAAAGTACCATCAAGACTGAGGCTGCTCTTCTACAACATGCACTTTGAGAAGTTCTGCATTGGAGATGCTCAAACATCTCAGTCACTAGTAGGAAAATGAAATGGTCC	AC154681|*Mus musculus* BAC clone RP24-314G22 from chromosome 14, complete sequence; Identities = 23/23 (100%)
1.5-M-5	**AGGGTT**AAGAACCTTTAGCTAGCATGGCGGCCGAAAAGAACCCGCTCCCCGCCTCCCAGGAGCTTCTGATTGGACAACCTG	AC166750| *Mus musculus* chromosome 8, clone RP23-9K18, complete sequence; Identities = 71/76 (93%)
1.5-M-6	**CTAGGG**TTAAGTTTACTCGGAATATTTCCAGGTCTCTCCT	AC161514| *Mus musculus* chromosome 7, clone RP23-378L12, complete sequence; Identities = 20/20 (100%)

^1^ Samples used for Splinkerette-PCR analysis were those sampled at 12 days (12-d) and 1.5 months (1.5-M) post-IPPGT. ^2^ TTAA consensus sequence is underlined. Sequences denoted as similar to TTAA are shadowed. Sequences corresponding to the end of PB (ITR) are shown by bold font. ^3^ Sequence homology analysis was performed using the BLASTN program set by DDBJ (http://blast.ddbj.nig.ac.jp/blastn?lang=ja).

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
