# Peer review of "In Vivo Piggybac-Based Gene Delivery towards Murine Pancreatic Parenchyma Confers Sustained Expression of Gene of Interest"

_ijms, 2019, doi:10.3390/ijms20133116_

Reviewer 1 Report

This manuscript describes the use of the piggyBac transposon system for in vivo gene delivery in the pancreas. In vivo electroporation is not a new technique, but the combination with transposon-based non-viral gene delivery and its application for the pancreas seems novel. The experiments are sound but I would like to bring up two points where the presentation could be improved:

1) Fig. 2 shows long-term fluorescence in animals that were co-injected with transposon and transposase. I think some statistics is missing here. How many animals were included in the experimental and the control groups? How reproducible was detection of fluorescence across the animal cohorts? What was the experimental variation of fluorescence between any two animals in quantitative terms? In order to appreciate the methods and its usefulness, some quantitative data need to be presented.

2) I would urge authors to include transposon sequence in Table 1, in other words the sequences should display the end of the transposon and the genomic flank.

Author Response

For reviewer 1;

Comments and Suggestions for Authors: This manuscript describes the use of the piggyBac transposon system for in vivo gene delivery in the pancreas. In vivo electroporation is not a new technique, but the combination with transposon-based non-viral gene delivery and its application for the pancreas seems novel. The experiments are sound but I would like to bring up two points where the presentation could be improved:

1) Fig. 2 shows long-term fluorescence in animals that were co-injected with transposon and transposase. I think some statistics is missing here. How many animals were included in the experimental and the control groups? How reproducible was detection of fluorescence across the animal cohorts? What was the experimental variation of fluorescence between any two animals in quantitative terms? In order to appreciate the methods and its usefulness, some quantitative data need to be presented.

Answer-1: Based on the reviewer’s comment, we checked the levels of tdTomato-derived fluorescence in the 3 intact samples for each group dissected at different days. The levels of mean fluorescence in each group were expressed as graphs (see right panel of Figure 2A in the revised text). Quantification of tdTomato-derived fluorescence in samples after IPPIGT demonstrated that there was a significant difference between the experimental and control groups in the levels of fluorescence when samples were dissected 1.5 months after IPPIGT (right panel of Figure 2A in the revised text). These are mentioned in the revised text (see L108-111, 133-136 and 342-353).  

2) I would urge authors to include transposon sequence in Table 1, in other words the sequences should display the end of the transposon and the genomic flank.

Answer-2: According to the reviewer’s suggestion, we placed some sequences of the end of the transposons linked to the genomic sequence (see Table 1 in the revised text; L213). 

Reviewer 2 Report

In this manuscript Sato et al. developed a piggyBac-based system to deliver genes of interest (GOI) into pancreas parenchyma, and showed stable expression of GOI up to 1.5 months after delivery. Data are nicely presented and convincing. However, several concerns need to be addressed:

First, longer time period than 1.5 months would be required to qualify as “long-term” integration.

Does electroporation cause damage to the pancreas? Previous studies showed electroporation of certain transposon system leads to pancreatic cancer. Did it occur with piggyBac?

What types of parenchymal cells were affected? Co-immunostaining of tdTomato with key pancreas cell identity markers is needed.

Ultimately, for such applications to be useful in clinical settings, Cas9 or shRNAs would be delivered to knock out/down genes, other than ectopically expressing or overexpressing genes as shown in this study. These would be nice to be included in the discussion.

Author Response

For reviewer 2;

Comments and Suggestions for Authors: In this manuscript Sato et al. developed a piggyBac-based system to deliver genes of interest (GOI) into pancreas parenchyma, and showed stable expression of GOI up to 1.5 months after delivery. Data are nicely presented and convincing. However, several concerns need to be addressed:

 1) First, longer time period than 1.5 months would be required to qualify as “long-term” integration.

Answer-1: This point the referee suggested was indeed not appropriate. Therefore, this phrase (long-term) was changed to the term “persistent” or “continuous” (see L90, 112, 164, 220, 237, 241 and 277 in the revised text).

 2) Does electroporation cause damage to the pancreas? Previous studies showed electroporation of certain transposon system leads to pancreatic cancer. Did it occur with piggyBac? 

Answer-2: We already noted that electroporation at higher stringency (i.e., over 40 voltage) caused deleterious effects (tissue chlorosis) on pancreatic cells from our previous experiments (Sato et al., Biotechnol. J. 2013, 8, 1355–1361). In this study, we used 20 voltage (as Tp) which has been proven to be less toxic to those cells (see L325 in the text). We never experienced occasional tumorigenesis in mice after IPPIGT with piggyBac vectors, as long as appropriate electroporation condition is applied to animals. 

 3) What types of parenchymal cells were affected? Co-immunostaining of tdTomato with key pancreas cell identity markers is needed.

Answer-3: Almost cells transfected by our IPPIGT are pancreatic acinar cells (Sato et al., Biotechnol. J. 2013, 8, 1355–1361). Only a few numbers of islets are transfected with this system. The idea (co-immunostaining of tdTomato with key pancreas cell identity markers is needed) that the referee suggests sounds nice. As for preferential expression of the exogenous DNA in a beta cell-specific manner, we are thinking of the use of insulin promoter upon IPPIGT. Our final goal is to induce in vivo transdifferntiation of acinar cells into the insulin-producing beta cells through direct gene delivery of factors (i.e., Pdx1) causing such event. In this context, preferential transfection of acinar cells may be convenient for our future experiment.       

 4) Ultimately, for such applications to be useful in clinical settings, Cas9 or shRNAs would be delivered to knock out/down genes, other than ectopically expressing or overexpressing genes as shown in this study. These would be nice to be included in the discussion.

Answer-4: As suggested by the referee, we added the new sentences (as shown below) to the revised text (L269-276): “Our IPPIGT system enables overexpression of a target gene in pancreatic cells (mainly acinar cells) in vivo. This technology is not only useful for persistent gene expression of a transgene when piggyBac gene delivery system is combined with IPPIGT (in this study), but also theoretically applicable to knock out or knock down of a target gene using a recently developed genome editing [as exemplified by clustered regularly interspaced short palindromic repeats/CRISPR associated proteins (CRISPR/Cas9)] and RNA interference (RNAi) technologies, respectively. These approaches will be valuable when IPPIGT is used for basic research towards future clinical applications.”